# Tear Film Dynamics between Low and High Contact Lens Dry Eye Disease Questionnaire (CLDEQ-8) Score with a Lehfilcon A Silicone Hydrogel Water Gradient Contact Lens: A Non-Invasive Methodology Approach

**DOI:** 10.3390/diagnostics13050939

**Published:** 2023-03-01

**Authors:** Raúl Capote-Puente, María-José Bautista-Llamas, José-María Sánchez-González

**Affiliations:** Department of Physics of Condensed Matter, Optica Area, Vision Research Group (CIVIUS), Faculty of Pharmacy, University of Seville, 41012 Seville, Spain

**Keywords:** tear film dynamics, contact lens dry eye questionnaire, CLDEQ-8, prelens tear film, lipid pattern, noninvasive break-up time, contact lens

## Abstract

The purpose of this study is to evaluate the tear film dynamics between individuals with low and high Contact Lens Dry Eye Disease Questionnaire (CLDEQ-8) scores when wearing Lehfilcon A silicone hydrogel water gradient contact lenses. In this study, we implemented a longitudinal, single-location, self-comparison investigation. Variables measured included conjunctival redness, lipid layer thickness, tear meniscus height, first and mean non-invasive break-up time, CLDEQ-8, and standard patient evaluation of eye dryness (SPEED). In the second phase, participants were re-evaluated after 30 days of wearing the contact lenses to assess the tear film wearing the lenses. In a longitudinal comparison by group, we found that lipid layer thickness decreased 1.52 ± 1.38 (*p* < 0.01) and 0.70 ± 1.30 (*p* = 0.01) Guillon patterns degrees in the low and high CLDEQ-8 group, respectively. MNIBUT increased in 11.93 ± 17.93 (*p* < 0.01) and 7.06 ± 12.07 (*p* < 0.01) seconds. Finally, LOT increased in 22.19 ± 27.57 (*p* < 0.01) and 16.87 ± 25.09 (*p* < 0.01). In conclusion, this study demonstrates the effectiveness of Lehfilcon A silicone hydrogel water gradient contact lenses in improving tear film stability and reducing subjective dry eye symptoms in individuals with low and high CLDEQ-8 scores. However, it also led to an increase in conjunctival redness and a decrease in tear meniscus height.

## 1. Introduction

The tear film is a thin layer of fluid that covers the preocular surface formed by cornea, bulbar, and palpebral conjunctiva, providing a smooth and clear surface for light to pass through [1]. The tear film is composed of three layers: the inner mucin layer, the middle aqueous layer, and the outer lipid layer. The inner mucin layer is produced by the goblet cells in the conjunctiva, a thin layer of tissue that lines the eyelids and covers the white part of the eye [2]. The mucin layer is secreted by goblet cells, Henle crypts, and Manz glands, being located in the deepest stratum of the lacrimal film containing glycoproteins [3]. The middle aqueous layer with a seromucosal composition is produced by the main lacrimal gland, located above the outer corner of each eye and the Krause and Wolfring accessory [4].

This layer contains water, electrolytes, urea, glucose, and several other molecules that supply the cornea with nutrients, including antimicrobials antibodies such as lysozyme or lactoferrin and enzymes that help to protect the eye from infection [5,6]. The aqueous layer also provides most of the tear film’s volume and is responsible for maintaining the tear film’s thickness. The outer lipid layer is produced by the meibomian glands, located in the eyelids. This layer helps to prevent the tear film from evaporating too quickly by creating a barrier on the surface of the eye [7]. The lipid layer also helps to spread the tear film evenly across the surface of the eye, which is important for maintaining a clear and smooth surface for light to pass through [1].

The tear film is constantly being replenished and refreshed, with new fluid being produced and old fluid being drained away through the tear ducts [8]. This constant turnover of fluid helps to ensure that the tear film remains fresh and healthy. The dynamics of the tear film are critical for maintaining the health and function of the eye. A disruption in the balance of any of the three layers is a hallmark about tear film instability that can lead to a variety of eye problems, such as dry eye syndrome, blepharitis, and other forms of ocular surface disorders [5]. Dry eye syndrome is a common condition characterized by a lack of sufficient tears to keep the eye lubricated. This can cause discomfort and a feeling of grittiness or burning in the eye [9]. Dry eye syndrome can be caused by a variety of factors, including age, certain medical conditions, and certain medications. Blepharitis is an inflammation of the eyelids that can cause redness, itching, and a feeling of grittiness or burning in the eye [8]. It can also cause the eyelids to become swollen and crusted. These disorders can lead to dry eye symptoms, tear film instability, and visual disturbances. The tear film is a complex and dynamic structure that plays a critical role in maintaining the health and function of the eye.

Contact lenses are a popular form of vision correction that are worn directly on the surface of the eye. They provide a convenient and discreet alternative to glasses and can be worn for extended periods of time [10]. However, wearing contact lenses can also lead to a variety of symptoms related to the health and comfort of the eye. One of the most common symptoms experienced by contact lens wearers is dry eye [11]. This can cause discomfort and a feeling of grittiness or burning in the eye [12]. Dry eye is particularly prevalent among contact lens wearers, with estimates suggesting that up to 30% of contact lens wearers experience dry eye symptoms [11].

To evaluate the symptoms of dry eye among contact lens wearers, a variety of questionnaires have been developed. One commonly used questionnaire is the Contact Lens Dry Eye Questionnaire (CLDEQ-8) [13]. This questionnaire is a validated tool that is designed to assess the symptoms of dry eye in contact lens wearers [13,14]. It consists of eight questions that assess symptoms such as dryness, burning, and discomfort. The questionnaire also includes a visual analogue scale (VAS) to rate the severity of symptoms [15]. The CLDEQ-8 questionnaire is a quick and easy tool that can be used to evaluate dry eye symptoms in contact lens wearers. It has been validated in multiple studies and has been shown to have good reliability and validity [16,17]. The questionnaire can be used to monitor the progression of dry eye symptoms over time, and to evaluate the effectiveness of interventions to improve symptoms [13]. In addition to dry eye symptoms, contact lens wearers may also experience other symptoms such as redness, itching, and foreign body sensation. These symptoms can be evaluated using questionnaires such as the Ocular Surface Disease Index (OSDI) and the Contact Lens Symptom Survey (CLSS) [18]. The use of questionnaires can provide valuable information on the prevalence, severity, and progression of symptoms among contact lens wearers, and can be used to evaluate the effectiveness of interventions to improve symptoms.

The dynamics of the tear film, such as its thickness and stability, can be affected by various factors, such as dry eye syndrome, contact lens wear, and aging [1]. Therefore, it is important to have accurate and non-invasive methods to measure the tear film dynamics in order to diagnose and monitor these conditions. One of the most commonly used non-invasive techniques for measuring the tear film dynamics is the non-invasive break-up time (NIBUT) test [19]. This test measures the time it takes for the tear film to break up after a blink, which reflects the stability of the tear film. The longer the NIBUT, the more stable the tear film. Another non-invasive method is the measurement of the tear film thickness using optical coherence tomography (OCT) [20]. This technique uses light waves to produce detailed images of the ocular surface, including the tear film. These images can be used to measure the thickness of the tear film and to detect any abnormalities.

Other non-invasive methods for measuring the tear film dynamics include the measurement of the tear meniscus height (TMH) [21]. The TMH test uses a digital camera to photograph the eye and measure the height of the tear meniscus, which reflects the volume of the tear film [21]. One of the most recent non-invasive methods for measuring the tear film dynamics is the use of interferometry [19]. This technique uses light waves to measure the thickness and stability of the tear film. The interferometer creates an interference pattern of the light reflecting off the tear film, which can be used to measure the thickness and stability of the tear film [19].

The purpose of this study is to evaluate the tear film dynamics between individuals with low and high Contact Lens Dry Eye Disease Questionnaire (CLDEQ-8) scores when wearing Lehfilcon A silicone hydrogel water gradient contact lenses. This study aims to use a non-invasive methodology approach to investigate the potential correlation between CLDEQ-8 scores and tear film dynamics. This research will provide insight into the relationship between dry eye symptoms and tear film dynamics in contact lens wearers, and may aid in the development of more effective interventions for managing dry eye symptoms in this population. Additionally, it will also provide information about the performance of Lehfilcon A silicone hydrogel water gradient contact lens in managing subjective dry eye symptoms by CLDEQ-8 questionnaire. The findings of this study may be useful for both clinicians and researchers in the field of ocular surface disorders and contact lens wear.

## 2. Materials and Methods

### 2.1. Design

In this study, we implemented a longitudinal, single-location, self-comparison investigation. It took place in the Optics and Optometry departments of the University of Seville’s Pharmacy School. The research was carried out in accordance with the guidelines set forth in the Helsinki Declaration and received approval from the University of Seville’s Ethical Committee Board (0384-N-22).

### 2.2. Subjects

All participants in the final analysis provided their informed consent and were provided with a detailed explanation of the study’s procedures. To be included in the study, participants had to meet the following criteria: (I) be in good ocular health and not currently undergoing any eye treatment, (II) be between the ages of 18 and 35, (III) score above 0 on the Contact Lens Dry Eye Questionnaire 8 (CLDEQ8), (IV) be daily or monthly silicone hydrogel contact lens wearers, and (V) have a spherical equivalent refraction of ≤5.50 diopters or less and refractive astigmatism of ≤1.50 diopters or less. Participants were excluded from the study if they had any of the following: (I) an active ocular infection or inflammation, or a history of ocular surgery, (II) were taking any medications that could affect the tear film or ocular surface, (III) had Sjogren syndrome, Rheumatoid arthritis, diabetes, or thyroid disorders, or (IV) were pregnant or breastfeeding.

### 2.3. Materials

All materials were described in a previous study [22]. Only listed below: Clinical Platform (ICP) Ocular Surface Analyzer (OSA) from SBM System^®^ (Orbassano, Torino, Italy), Nonmydriatic infrared meibography digital fundus camera Cobra^®^ HD (Construzione Strumenti Oftalmici CSO^®^, Firenze, Italy), Contact Lens Dry Eye Questionnaire 8 (CLDEQ-8) [13,14] and the Standard Patient Evaluation of Eye Dryness (SPEED). Silicone hydrogel contact lens (TOTAL 30^®^, Alcon Inc., Fort Worth, TX, USA) and a multipurpose solution (MPS) (Lens 55^®^ Care Hyaluropolimer Plus 360 mL, Servilens Fit and Cover^®^, Granada, Spain) for all subjects.

### 2.4. Procedure

In the initial phase, participants were selected based on specific criteria and their samples were collected from the non-optometry field. After this period, surveys and non-invasive tests were conducted to assess tear film fluctuations. Variables measured included conjunctival redness, lipid layer thickness, tear meniscus height, and meibomian gland dysfunction. In the second phase, participants were re-evaluated with the same variables measured in phase one after 30 days of wearing the contact lenses to assess the tear film in front of the lenses. Participants were instructed to follow a specific lens care regimen and avoid eye drops or lubricants. The test conditions were consistent, and the measurements were alternated between eyes. Participants were also instructed to blink normally between measurements.

### 2.5. Statistical Analysis

Data analysis was conducted using IBM Corp’s SPSS software (version 26.0). Descriptive statistics such as mean and standard deviation were used. The normality of the data was determined using the Shapiro-Wilk test. Differences in categorical variables were assessed using the chi-square test, and differences in numerical variables between different time points were evaluated using the Wilcoxon test. The group were separate with the CLDEQ-8 diagnostic criteria established by Chalmers et al. [13,14] of 12 score points. Low CLDEQ-8 group were patients with a baseline CLDEQ-8 ≤ 12 and high CLDEQ-8 group were patients with a baseline CLDEQ-8 > 12 score points. All tests were set at a significance level of 95% (*p* value < 0.05). The sample size was calculated using the GRANMO^®^ calculator, with a two-tailed test, alpha and beta risks of 5% and 20%, respectively, and an estimated standard deviation of 0.45. The recommended sample size was 28 subjects.

## 3. Results

A group of thirty-one subjects with low levels of astigmatism and myopia were fitted with silicone hydrogel contact lenses (Lehfilcon A). From the sample, 7 (22.6%) were male and 24 (77.4%) were female. Twenty-one subjects were from Italy (67.75%), and the rest of the patients were from different countries including Spain (12.90%), Mexico (6.46%), Slovenia (3.22%), Poland (3.22%), Germany (3.22%), and Austria (3.22%). Mean age of the subjects was 22.23 ± 1.39 (19 to 25) years old. The refraction of the subjects was sphere (diopters) −2.64 ± 1.15 (−5.50 to −0.50), cylinder (diopters) −0.44 ± 0.37 (−1.50 to 0.00), and axis (degrees, °) 111.44 ± 70.08 (5.00 to 180.00).

Regarding visual acuity (Log MAR): −0.03 ± 0.05 (−0.10 to 0.10). Mean cornea keratometry were flat corneal meridian (mm) 7.87 ± 0.31 (7.40 to 8.74), steep corneal meridian (mm) 7.73 ± 0.29 (7.25 to 8.61), and mean corneal meridian (mm) 7.80 ± 0.30 (7.37 to 8.67). Meibomian gland dysfunction (MGD) was studied along the percentage of loss. Superior eyelid MGD (%) 28.87 ± 15.11 (10.30 to 96.20) and inferior eyelid MGD (%) 49.69 ± 17.86 (17.00 to 87.30). Finally, contact lens power (diopters) was −2.56 ± 1.12 (−5.00 to −0.75).

Baseline measurements divided within the contact lens subjective questionnaire groups (Low CLDEQ-8 and high CLDEQ-8) were presented in Table 1. In the same term, one-month measurements divided within the contact lens subjective questionnaire groups (Low CLDEQ-8 and high CLDEQ-8) were presented in Table 2. All the patients were able to comfortably wear the contact lenses for 30 days, with the exception of one person who experienced a minor irritation that cleared up after a few days. The contact lenses were worn for an average of 5.61 days per week, 8.95 h per day, and 3.68 h on the one-month follow-up visit. Pyramid graph about baseline and one-month conjunctival redness classification, lipid layer thickness interferometry, tear meniscus height, and first NIBUT were presented in Figure 1. Pyramid graph about baseline and one-month mean NIBUT, lid opening time, SPEED, and CLDEQ-8 were presented in Figure 2.

In a longitudinal comparison by group, we found that conjunctival redness classification change from baseline to one-month 0.00 ± 0.75 (*p* = 0.99) and increase 0.33 ± 0.56 (*p* = 0.01) degrees in the low and high CLDEQ8 group, respectively. Lipid layer thickness decreased 1.52 ± 1.38 (*p* < 0.01) and 0.70 ± 1.30 (*p* = 0.01) Guillon patterns degrees in the low and high CLDEQ-8 group, respectively. Meanwhile TMH remain stable between groups but decreased 0.07 ± 0.05 (*p* < 0.01) and 0.05 ± 0.05 mm (*p* < 0.01) in the low and high CLDEQ-8 group, respectively. Regarding the stability of the tear film, FNIBUT decreased 0.33 ± 1.66 (*p* < 0.01) and 0.29 ± 1.46 (*p* < 0.01) seconds, MNIBUT increased in 11.93 ± 17.93 (*p* < 0.01) and 7.06 ± 12.07 (*p* < 0.01) seconds, and finally LOT increased in 22.19 ± 27.57 (*p* < 0.01) and 16.87 ± 25.09 (*p* < 0.01). CLDEQ-8 increased 0.72 ± 7.58 (*p* = 0.71) points and decreased 2.83 ± 11.13 (*p* = 0.37) points, respectively. SPEED test decreased 1.38 ± 5.31 (*p* = 0.01) and 2.50 ± 7.51 (*p* = 0.01) points, respectively.

## 4. Discussion

### 4.1. Summary of Findings

Our findings could be summarized comparing the two groups, the following changes were observed: conjunctival redness increased in high CLDEQ-8 group. Lipid layer decreased in both groups, especially in low CLDEQ-8. TMHs remain stable. FNIBUT decreased in both groups without statistically signification. MNIBUT and LOT achieved relevant and significantly increased in both groups. Finally, subjective dry eye disease sensation questionnaire specially decreased in high CLDEQ-8 group.

### 4.2. Comparison with Other Authors Outcomes

The daily version of Lehfilcon A has been previously studied. Marx et al. [23] conducted to determine the level of satisfaction and comfort among first-time contact lens users who were given Delefilcon A (DAILIES TOTAL1) daily disposable lenses. The study was conducted in various European locations and spanned a period of two weeks. Participants were initially fitted with Delefilcon A lenses and then evaluated at the beginning of the trial and after the first and second weeks. The results revealed that the study lenses were found to have a higher mean score for subject-reported quality of vision and comfort compared to their traditional glasses. Over 90% of the subjects stated that the lenses were more comfortable than they had anticipated and 92% expressed an interest in purchasing them. The investigators found that the fit of the study lenses was acceptable in at least 97% of the subjects. The study ultimately concluded that practitioners could expect positive results when transitioning first-time contact lens wearers from glasses to Delefilcon A daily disposable contact lenses.

Szczesna-Iskander et al. [24] conducted to evaluate the pre-lens tear film surface quality (TFSQ) of a new water gradient silicone hydrogel material that is used in daily disposable lenses, in comparison to another daily disposable lens from the same manufacturer. Eleven subjects participated in the study and wore two different types of lenses for two non-consecutive days. The TFSQ was measured using non-invasive interferometry and subjective comfort was also assessed. The results showed that both lenses resulted in a reduction of TFSQ compared to the bare eye condition. The new water gradient silicone hydrogel material had a statistically significant smaller impact on TFSQ than the high-water content material. The measurement methodology was found to have a high level of linearity with respect to the lens material and there was a statistically significant correlation between the TFSQ results of the two lenses. However, the correlation between subjective comfort and the lenses was found to be insignificant. This research supports the minimal effect on the tear meniscus height of Delefilcon A.

In a similar line of research Fujimoto et al. [25] aimed to determine if the water gradient technology of Delefilcon A-based soft contact lenses improves tear film dynamics. The study was observational and retrospective and included 50 asymptomatic users of Delefilcon A or Narafilcon A SCLs. The study measured the thin aqueous layer break, non-invasive tear break-up time, tear meniscus height, subjective dryness, and higher-order aberrations. The measurements were taken at three visits: the first with the bare eye, the second with the SCL-worn eye after 15 min, and the third 30 ± 5 days after the second visit after the SCL was worn for at least 5 h. The study found that the water gradient technology of Delefilcon A reduced thin aqueous layer break and increased non-invasive tear break-up time. Additionally, it reduced tear meniscus height and total ocular higher-order aberrations, and improved lens performance.

Lipid layer thickness reduction was determined by the presence of the biomimetic content on the surface of the contact lens. Mao et al. [26] observed that surfaces with high-aspect ratio nanostructure have the ability to kill bacteria. The physical interaction between the nanostructure and the cell membrane causes the cells to break apart. Recent research has been able to transfer this ability to artificial surfaces. However, these surfaces may have different properties compared to those found in nature. The review looks at recent progress in developing bactericidal surfaces and analyses the factors that influence their effectiveness and the mechanism behind the cell rupture. It uses a holistic approach, combining different factors such as interaction mechanisms, material properties, and fabrication techniques, to understand the effect of surface topography and its potential use in soft contact lenses.

Wesley et al. [27] investigated Lehfilcon A, but applied a subjective methodology in the wettability measurements. Some 115 subjects completed the study and were divided into two groups: one wearing the investigational Delefilcon A lens and the other wearing the control Comfilcon A lens, both for 3 months. The study measured distance visual acuity, lens fit and movement, centration, front and back surface deposits, and front surface wettability. The results showed that both lenses provided excellent visual acuity, optimal lens fitting characteristics, a clean surface, high wettability, and no ocular adverse events after 3 months of lens wear. The Delefilcon A lens showed better performance in terms of centration and lens fit/movement than the Comfilcon A lens.

### 4.3. Strengths and Limitations

The strengths of the current study include the use of a non-invasive methodology approach to assess tear film dynamics in individuals with varying levels of contact lens dry eye disease (CLDEQ-8 scores) using a Lehfilcon A silicone hydrogel water gradient contact lens. This approach allowed for the objective measurement of tear film parameters without the need for invasive methods. Additionally, the use of the CLDEQ-8 questionnaire provided a standardized and validated method for assessing dry eye symptoms in individuals wearing contact lenses.

Certainly, using a non-invasive technology to measure tear film dynamics provides several advantages over traditional questionnaires. Firstly, non-invasive methods, such as the ones used in our study, do not require physical contact with the eye, making them more comfortable for the patient and reducing the risk of infection or injury. This is particularly important in individuals with dry eye disease, as their eyes may be more sensitive to physical contact [28].

Secondly, non-invasive methods can provide more objective and quantitative data compared to subjective questionnaires. By directly measuring tear film parameters such as lipid layer thickness, tear meniscus height, and non-invasive break-up time, we can obtain accurate and precise measurements that can be used to monitor changes in tear film dynamics over time [21].

Finally, non-invasive methods are less dependent on patient interpretation and recall, which can be a source of bias in questionnaires. This makes non-invasive methods more reliable and reproducible, which is important for ensuring the accuracy and validity of research findings [29].

By highlighting these advantages, we can better demonstrate the innovation of our study and the importance of using non-invasive methods to assess tear film dynamics. Comparing our non-invasive methods with other questionnaires can help to further emphasize the unique contribution of our study to the field of dry eye disease research.

The sample size was also appropriate and calculated using an appropriate sample size calculator. The study also used multiple measurements to evaluate the tear film dynamics, which provide a more comprehensive assessment of the tear film. Furthermore, the use of a silicone hydrogel water gradient contact lens is a relevant and clinically significant lens type, which is widely used today. Finally, the study compared the tear film dynamics between low and high CLDEQ-8 score groups, which is an important aspect to understand the impact of dry eye symptoms on tear film parameters.

One limitation of the current study is that it only included a single type of contact lens, the Lehfilcon A silicone hydrogel water gradient contact lens. While this lens type is widely used and clinically relevant, it may not be representative of all contact lens types and the findings may not be generalizable to other lens types. Additionally, the study only included a single follow-up time point of 30 days, which may not capture the full extent of tear film dynamics over a longer period of time. Furthermore, the study was conducted on a small sample size which may not be representative of the entire population. Another limitation is that the study did not include a control group of individuals not wearing contact lenses, thus it is not possible to determine if the tear film dynamics observed are specific to contact lens wear or if they would have been observed in non-lens wearers as well. Additionally, the study only measured tear film dynamics at specific time points, it is not possible to understand the tear film dynamics over time, and also, the study did not include any subjective measurements of dry eye symptoms, it would have been beneficial to have both objective and subjective measures to provide a more complete understanding of the relationship between contact lens wear and dry eye symptoms.

Moreover, the study only used a single non-invasive method, Ocular Surface Analyzer (OSA) and Meibographer, to assess tear film dynamics. Other methods such as the Interferometer or Schirmer test may have provided additional information about tear film dynamics. Also, the study did not include any measurement of the lens surface, which could have provided information about the lens surface and how it contributes to the tear film dynamics. In addition, the study did not include any measure of the lens movement during the blink, this would have provided important information about the lens–eyelid interaction and how it affects the tear film dynamics. Finally, the study did not include any measure of the tear film break-up time, which is an important measure of tear film stability and dry eye. The current study provides important information about tear film dynamics in individuals wearing a specific type of contact lens, but additional studies with larger sample sizes and longer follow-up periods, including a control group and other lens types, and using a combination of objective and subjective measures, and other non-invasive methods, are needed to fully understand the relationship between contact lens wear and tear film dynamics.

### 4.4. Future Lines of Research

Future research on this topic could explore the potential relationship between the severity of dry eye symptoms as measured by the CLDEQ-8 questionnaire and the tear film parameters measured by the non-invasive methods. This could provide insight into which specific tear film parameters are most affected by dry eye symptoms and could potentially be used as biomarkers for dry eye in contact lens wearers. Another line of research could be to investigate the impact of various lens care solutions on the tear film dynamics. Different lens care solutions have different compositions, and some may have a greater impact on the tear film than others. Additionally, the impact of the lens replacement schedule (daily, weekly, monthly) on the tear film dynamics could also be studied.

It would also be interesting to explore the relationship between the tear film dynamics and the lens design. For example, different lens materials, designs, and geometries could affect the tear film dynamics differently. This could be relevant in the development of new contact lenses with improved comfort and performance. Additionally objective tear film dynamics measurements could be added as future research. TF-OSI dynamic objective scattering index of tear film provide a more complete understanding of the effect of contact lens wear on tear film stability. Finally, future research could include a greater diversity of subjects by including different age groups, ethnicities, and genders to better understand the tear film dynamics in different populations. This would allow the generalization of the findings to a larger population and improvement of the clinical significance of the results.

## 5. Conclusions

This study demonstrates the effectiveness of Lehfilcon A silicone hydrogel water gradient contact lenses in improving tear film stability and reducing subjective dry eye symptoms in individuals with low and high CLDEQ-8 scores. However, it also led to an increase in conjunctival redness and a decrease in tear meniscus height. CLDEQ-8 questionnaire provides a standardized and quantitative measure of dry eye symptoms in contact lens wearers.

Water gradient contact lenses present a promising option for patients who experience dry eye symptoms related to contact lens wear. They have been shown to improve objective measures such as artificial tear film dynamics and subjective measures such as dry eye questionnaire scores. However, it is important to have close supervision of the ocular surface to monitor and control parameters such as ocular redness and tear layer volume.

## Figures and Tables

**Figure 1 diagnostics-13-00939-f001:**
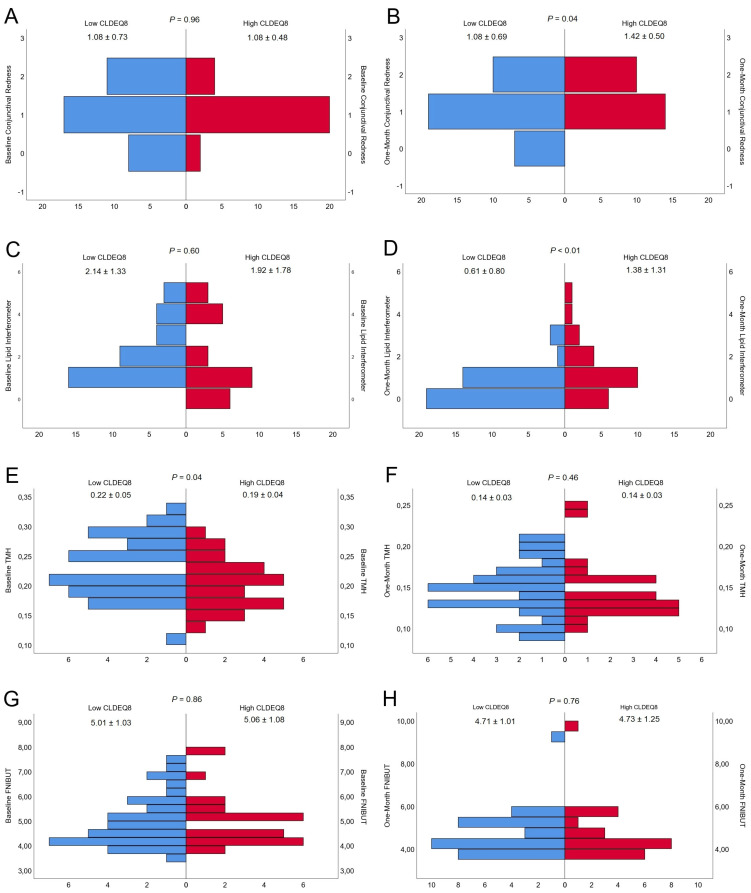
Baseline and One Month Comparison of (**A**,**B**) Conjunctival Redness, (**C**,**D**) Lipid Layer Interferometry, (**E**,**F**) Tear Meniscus Height (TMH), and (**G**,**H**) First Non-Invasive Break Up Time (FNIBUT). The figure consists of eight pyramid-shaped graphs, with four left representing the baseline measurements and four right representing the measurements taken one month later.

**Figure 2 diagnostics-13-00939-f002:**
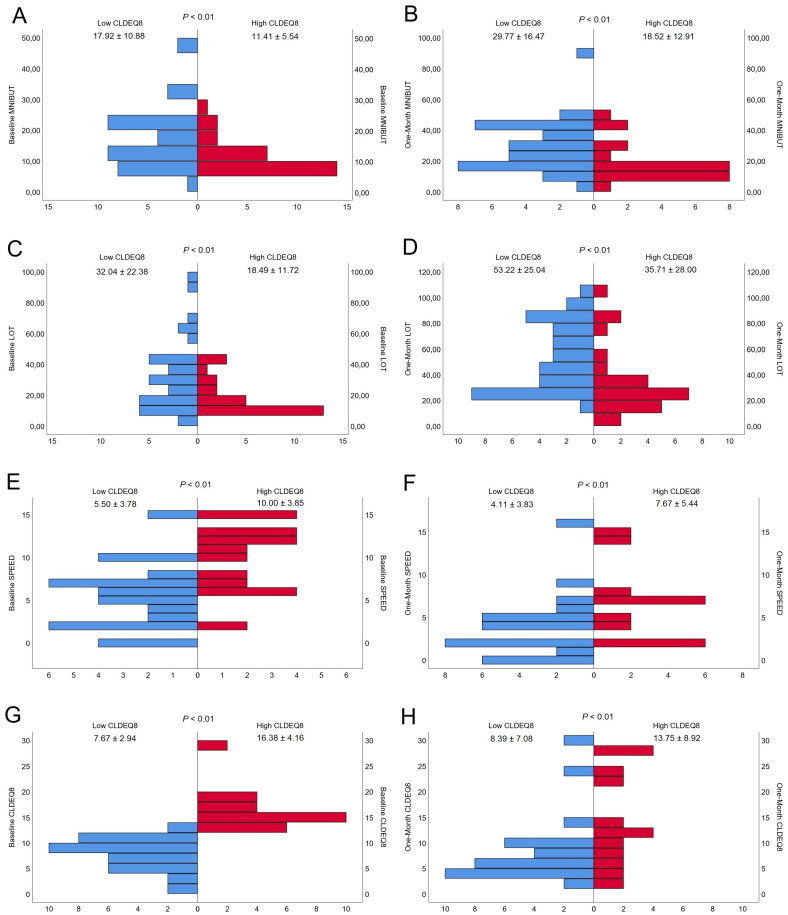
Baseline and One Month Comparison of (**A**,**B**) Mean Non-invasive Break-up Time, (**C**,**D**) Lid Opening Time (LOT), (**E**,**F**) Standard Patient Evaluation of Eye Dryness (SPEED) and (**G**,**H**) Contact Lens Dry Eye Questionnaire (CLDEQ-8). The figure consists of eight pyramid-shaped graphs, with four left representing the baseline measurements and four right representing the measurements taken one month later.

**Table 1 diagnostics-13-00939-t001:** Baseline outcomes divided by CLDEQ-8 group.

Variable	Low CLDEQ-8	High CLDEQ-8	*p* Value
Conjunctival Redness Classification (Efron Scale)	1.08 ± 0.73 (0.00 to 2.00)	1.08 ± 0.48 (0.00 to 2.00)	0.96
Lipid Layer Thickness Interferometry (Guillon Pattern)	2.14 ± 1.33 (1.00 to 5.00)	1.92 ± 1.78 (0.00 to 5.00)	0.60
Tear Meniscus Height (Millimeters)	0.22 ± 0.05 (0.11 to 0.32)	0.19 ± 0.04 (0.13 to 0.29)	0.04 *
First NIBUT (seconds)	5.01 ± 1.03 (3.60 to 7.36)	5.06 ± 1.08 (3.92 to 7.80)	0.86
Mean NIBUT (seconds)	17.92 ± 10.88 (4.50 to 49.76)	11.41 ± 5.54 (6.02 to 25.14)	<0.01 *
Lid Opening Time (seconds)	32.04 ± 22.38 (5.04 to 93.60)	18.49 ± 11.72 (7.76 to 46.32)	<0.01 *
CLDEQ8 (Score Points)	7.67 ± 2.94 (1.00 to 12.00)	16.38 ± 4.16 (13.00 to 29.00)	<0.01 *
SPEED Test (Score Points)	5.50 ± 3.78 (0.00 to 15.00)	10.00 ± 3.85 (2.00 to 15.00)	<0.01 *

NIBUT: Non-Invasive Break Up Time, CLDEQ-8: Contact Lens Dry Eye Questionnaire, SPEED: Standard Patient Evaluation of Eye Dryness. * Statistically significant within U of Mann Whitney.

**Table 2 diagnostics-13-00939-t002:** One-month outcomes divided by CLDEQ-8 group.

Variable	Low CLDEQ-8	High CLDEQ-8	*p*-Value
Conjunctival Redness Classification (Efron Scale)	1.08 ± 0.69 (0.00 to 2.00)	1.42 ± 0.50 (1.00 to 2.00)	0.04 *
Lipid Layer Thickness Interferometry (Guillon Pattern)	0.61 ± 0.80 (0.00 to 3.00)	1.38 ± 1.31 (0.00 to 5.00)	<0.01 *
Tear Meniscus Height (millimeters)	0.14 ± 0.03 (0.09 to 0.21)	0.14 ± 0.03 (0.10 to 0.25)	0.46
First NIBUT (seconds)	4.71 ± 1.01 (3.60 to 9.00)	4.73 ± 1.25 (3.60 to 9.56)	0.76
Mean NIBUT (seconds)	29.77 ± 16.47 (5.50 to 91.14)	18.52 ± 12.91 (5.72 to 46.72)	<0.01 *
Lid Opening Time (seconds)	53.22 ± 25.04 (18.56 to 103.71)	35.71 ± 28.00 (7.52 to 106.40)	<0.01 *
CLDEQ8 (Score Points)	8.39 ± 7.08 (2.00 to 29.00)	13.75 ± 8.92 (2.00 to 28.00)	<0.01 *
SPEED Test (Score Points)	4.11 ± 3.83 (0.00 to 16.00)	7.67 ± 5.44 (2.00 to 19.00)	<0.01 *

NIBUT: Non-Invasive Break Up Time, CLDEQ-8: Contact Lens Dry Eye Questionnaire, SPEED: Standard Patient Evaluation of Eye Dryness. * Statistically significant within U of Mann Whitney.

## Data Availability

The data presented in this study are available on request from the corresponding author. The data are not publicly available due to their containing information that could compromise the privacy of research participants.

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
