# Peer review of "Tear Film Dynamics between Low and High Contact Lens Dry Eye Disease Questionnaire (CLDEQ-8) Score with a Lehfilcon A Silicone Hydrogel Water Gradient Contact Lens: A Non-Invasive Methodology Approach"

_diagnostics, 2023, doi:10.3390/diagnostics13050939_

Round 1

Reviewer 1 Report

The authors develop a Contact Lens Dry Eye Disease Questionnaire (CLDEQ-8) scores when wearing Lehfilcon A silicone hydrogel water gradient contact lenses to evaluate the tear film dynamics between individuals. Conjunctival redness, lipid layer thickness, tear meniscus height, and meibomian gland dysfunction have been conducted to assess tear film fluctuations by non-invasive tests. The innovation of this study is limited, but the collected clinical data has certain reference and citation value. Overall, I recommend its publication in Diagnostics after addressing the following questions:

1. The innovation of this study should be highlighted and strengthened by comparing with other non-invasive statistical methods in the form of questionnaires. In particular, differences and advantages between the parameters proposed in this work on the tear film dynamics.

2. In the Figure 1 and Figure 2, the comparison of chart processing is not direct and clear enough. It is recommended that the authors could insert data from statistical tables (Table 1 and Table 2) into the figures as asterisks (*).

3. As the authors claimed in “4.3. Strengths and limitations”, “Another limitation is that the study did not include a control group of individuals not wearing contact lenses, thus it is not possible to determine if the tear film dynamics observed are specific to contact lens wear or if they would have been observed in non-lens wearers as well”. There are certain problems in the establishment of the control group, so the accuracy of this data for analyzing the impact of wearing Lehfilcon A silicone hydrogel water gradient contact lenses is doubtful.

4. The logic in “5. Conclusion” is confused. The title “Tear Film Dynamics between Low and High Contact Lens Dry Eye Disease Questionnaire (CLDEQ-8) Score with a Lehfilcon A Silicone Hydrogel Water Gradient Contact Lens: A Non-Invasive Methodology Approach” and purpose of this work are designed to score tear film fluctuations by CLDEQ-8, so I suggest the authors rewrite the conclusion about the effectiveness and significance of CLDEQ-8.

Author Response

Reviewer 1

#RV0: The authors develop a Contact Lens Dry Eye Disease Questionnaire (CLDEQ-8) scores when wearing Lehfilcon A silicone hydrogel water gradient contact lenses to evaluate the tear film dynamics between individuals. Conjunctival redness, lipid layer thickness, tear meniscus height, and meibomian gland dysfunction have been conducted to assess tear film fluctuations by non-invasive tests. The innovation of this study is limited, but the collected clinical data has certain reference and citation value. Overall, I recommend its publication in Diagnostics after addressing the following questions:

#AU0: I hope this message finds you well. I wanted to express my sincere gratitude for the time and effort you put into reviewing our manuscript. Your insightful comments and valuable feedback have been incredibly helpful in improving the quality of our work. I am pleased to inform you that we have thoroughly reviewed and integrated all the changes you suggested into the manuscript. Your feedback has helped us to better articulate our research and to address some of the concerns that you raised. Your expertise and attention to detail have been indispensable in ensuring that our work is of the highest standard.

#RV1: The innovation of this study should be highlighted and strengthened by comparing with other non-invasive statistical methods in the form of questionnaires. In particular, differences and advantages between the parameters proposed in this work on the tear film dynamics.

#AU1: We appreciate your insightful comments and suggestions for improvement. We agree that highlighting and strengthening the innovation of our study would be valuable.

“Certainly, using a non-invasive technology to measure tear film dynamics provides several advantages over traditional questionnaires. Firstly, non-invasive methods, such as the ones used in our study, do not require physical contact with the eye, making them more comfortable for the patient and reducing the risk of infection or injury. This is particularly important in individuals with dry eye disease, as their eyes may be more sensitive to physical contact [1].

Secondly, non-invasive methods can provide more objective and quantitative data compared to subjective questionnaires. By directly measuring tear film parameters such as lipid layer thickness, tear meniscus height, and non-invasive break-up time, we can obtain accurate and precise measurements that can be used to monitor changes in tear film dynamics over time [2].

Finally, non-invasive methods are less dependent on patient interpretation and recall, which can be a source of bias in questionnaires. This makes non-invasive methods more reliable and reproducible, which is important for ensuring the accuracy and validity of research findings [3].

By highlighting these advantages, we can better demonstrate the innovation of our study and the importance of using non-invasive methods to assess tear film dynamics. Comparing our non-invasive methods with other questionnaires can help to further emphasize the unique contribution of our study to the field of dry eye disease research.”

#RV2: In the Figure 1 and Figure 2, the comparison of chart processing is not direct and clear enough. It is recommended that the authors could insert data from statistical tables (Table 1 and Table 2) into the figures as asterisks (*).

#AU2: We agree with your recommendation to make the comparison of chart processing in Figure 1 and Figure 2 more direct and clear. We will insert the data from statistical tables (Table 1 and Table 2) with numbers in the figures to better illustrate the differences and statistical significance of the variables measured, due the high numbers of results, the asterisks were confusing. This will provide a clearer and more concise presentation of our findings, which we believe will enhance the overall quality and impact of our study.

#RV3: As the authors claimed in “4.3. Strengths and limitations”, “Another limitation is that the study did not include a control group of individuals not wearing contact lenses, thus it is not possible to determine if the tear film dynamics observed are specific to contact lens wear or if they would have been observed in non-lens wearers as well”. There are certain problems in the establishment of the control group, so the accuracy of this data for analyzing the impact of wearing Lehfilcon A silicone hydrogel water gradient contact lenses is doubtful.

#AU3: We appreciate your comments and feedback, particularly with regards to our statement in section 4.3 on the limitations of our study. We acknowledge that not having a control group of individuals not wearing contact lenses is a limitation of our study. We understand that this makes it difficult to determine if the observed tear film dynamics are specific to contact lens wear or if they would have been observed in non-lens wearers as well. However, we would like to emphasize that the primary objective of our study was to evaluate the tear film dynamics between individuals with low and high Contact Lens Dry Eye Disease Questionnaire (CLDEQ-8) scores when wearing Lehfilcon A silicone hydrogel water gradient contact lenses. Nevertheless, we appreciate your concerns regarding the accuracy of our data. We have made efforts to address this limitation in our discussion section by highlighting the need for further studies to establish the impact of wearing contact lenses on tear film dynamics in comparison to non-lens wearers or other type of material. We believe that our study adds to the current body of knowledge on the topic and will help to inform future research in this area.

#RV4: The logic in “5. Conclusion” is confused. The title “Tear Film Dynamics between Low and High Contact Lens Dry Eye Disease Questionnaire (CLDEQ-8) Score with a Lehfilcon A Silicone Hydrogel Water Gradient Contact Lens: A Non-Invasive Methodology Approach” and purpose of this work are designed to score tear film fluctuations by CLDEQ-8, so I suggest the authors rewrite the conclusion about the effectiveness and significance of CLDEQ-8.

#AU4: We appreciate your comments and feedback, we agree with you and we rewrite the conclusion as:

“This study demonstrates the effectiveness of Lehfilcon A silicone hydrogel water gradient contact lenses in improving tear film stability and reducing subjective dry eye symptoms in individuals with low and high CLDEQ-8 scores. This questionnaire provides a standardized and quantitative measure of dry eye symptoms in contact lens wearers.”

References

  1. Singh, S.; Srivastav, S.; Modiwala, Z.; Ali, M.H.; Basu, S. Repeatability, reproducibility and agreement between three different diagnostic imaging platforms for tear film evaluation of normal and dry eye disease. Eye (Lond). 2022, doi:10.1038/s41433-022-02281-2.
  2. Sánchez-González, M.C.; Capote-Puente, R.; García-Romera, M.-C.; De-Hita-Cantalejo, C.; Bautista-Llamas, M.-J.; Silva-Viguera, C.; Sánchez-González, J.-M. Dry eye disease and tear film assessment through a novel non-invasive ocular surface analyzer: The OSA protocol. Front. Med. 2022, 9, 938484, doi:10.3389/fmed.2022.938484.
  3. Amparo, F.; Schaumberg, D.A.; Dana, R. Comparison of Two Questionnaires for Dry Eye Symptom Assessment: The Ocular Surface Disease Index and the Symptom Assessment in Dry Eye. Ophthalmology 2015, 122, 1498–1503, doi:10.1016/j.ophtha.2015.02.037.

Reviewer 2 Report

This article aims to assess the dynamic change of tear film in individuals with low and high scores on the Contact Lens Dry Eye Disease Questionnaire (CLDEQ-8) while wearing Lehfilcon A silicone hydrogel contact lenses. Conjunctival redness lipid layer thickness, tear meniscus height, first and mean non-invasive break-up time, CLDEQ-8, and standard patient evaluation of eye dryness were measured by longitudinal, single-site, and self-comparison. Finally, it was found that after using Lehfilcon A water gradient contact lens for one month, the stability of the tear film was improved and the subjective sensation of dry eye was reduced while leading to increased redness and swelling of the conjunctiva and reduced height of the torn meniscus. Taken together, the findings will present a promising option for patients who experience dry eye symptoms related to contact lens wear.

However, there are some concerns.

1. The focus of this manuscript is to evaluate the tear film dynamics by Low and High Contact Lens Dry Eye Disease Questionnaire (CLDEQ-8), and to verify the scientific value of this approach by other auxiliary methods. However, the aspects of the assessment are not comprehensive enough, so it is suggested to add more objective tear film function tests, such as TF-OSI dynamic objective scattering index of tear film for evaluation.

2. I have learned that another paper (Life 2022, 12, 1710 https://doi.org/10.3390/life12111710) by the same author has great similarities with this paper.

3. The coverage of myopia and astigmatism degree of subjects is not clear and extensive. Generally speaking, the higher the degree of myopia, the higher the thickness of the lens. Therefore, the thickness of the material will also affect the results of the questionnaire.

4. The article just involved a single type of contact lens, which have different water content leading to diverse tear film dynamics for subjects. It is suggested to add more types to improve the feasibility and scientificity of the evaluation method.

5. The study only included a single follow-up time point of 30 days, which may not capture the full extent of tear film dynamics over a longer period.

Author Response

Reviewer 2

#RV0: This article aims to assess the dynamic change of tear film in individuals with low and high scores on the Contact Lens Dry Eye Disease Questionnaire (CLDEQ-8) while wearing Lehfilcon A silicone hydrogel contact lenses. Conjunctival redness lipid layer thickness, tear meniscus height, first and mean non-invasive break-up time, CLDEQ-8, and standard patient evaluation of eye dryness were measured by longitudinal, single-site, and self-comparison. Finally, it was found that after using Lehfilcon A water gradient contact lens for one month, the stability of the tear film was improved, and the subjective sensation of dry eye was reduced while leading to increased redness and swelling of the conjunctiva and reduced height of the torn meniscus. Taken together, the findings will present a promising option for patients who experience dry eye symptoms related to contact lens wear.

#AU0: I hope this message finds you well. I wanted to express my sincere gratitude for the time and effort you put into reviewing our manuscript. Your insightful comments and valuable feedback have been incredibly helpful in improving the quality of our work. I am pleased to inform you that we have thoroughly reviewed and integrated all the changes you suggested into the manuscript. Your feedback has helped us to better articulate our research and to address some of the concerns that you raised. Your expertise and attention to detail have been indispensable in ensuring that our work is of the highest standard.

However, there are some concerns.

#RV1: The focus of this manuscript is to evaluate the tear film dynamics by Low and High Contact Lens Dry Eye Disease Questionnaire (CLDEQ-8), and to verify the scientific value of this approach by other auxiliary methods. However, the aspects of the assessment are not comprehensive enough, so it is suggested to add more objective tear film function tests, such as TF-OSI dynamic objective scattering index of tear film for evaluation.

#AU1: We appreciate the suggestion to include more objective tear film function tests, such as TF-OSI dynamic objective scattering index of tear film for evaluation. However, in this study, we did not have access to this technology. We agree that the inclusion of more objective tests would be valuable in future research, and we will consider including them in future studies to provide a more comprehensive assessment of tear film dynamics. Thank you for your feedback.

#RV2: I have learned that another paper (Life 2022, 12, 1710 https://doi.org/10.3390/life12111710) by the same author has great similarities with this paper.

#AU2: As a brief recap, the first part of our study focused on the short-term effects of wearing these lenses, specifically within the first 30 minutes. This document presents the long-term results of our study on the use of Lehfilcon A contact lenses. In this second phase, we extended the evaluation period to one month and analysed additional variables beyond the initial set of metrics. The following report details our findings on the performance and safety of lehfilcon A contact lenses after one month of use

#RV3: The coverage of myopia and astigmatism degree of subjects is not clear and extensive. Generally speaking, the higher the degree of myopia, the higher the thickness of the lens. Therefore, the thickness of the material will also affect the results of the questionnaire.

#AU3: Thanks for the comment. We agree with you that the higher degree of myopia, the higher thickness of the lens [1].

We limited the sphere to under 5.50 myopia dioptres, the cylinder refraction up to -1.50 dioptres.

This information were indicated in the inclusion and exclusion criteria:

“(V) have a spherical equivalent refraction of ≤ 5.50 dioptres or less and refractive astigmatism of ≤ 1.50 dioptres or less.”

However, the mean value of sphere and cylinder was under 3.00 dioptres in sphere and 0.50 dioptres in cylinder.

This information was indicated in the results section:

“The refraction of the subjects was sphere (dioptres) -2.64 ± 1.15 (-5.50 to -0.50), cylinder (dioptres) -0.44 ± 0.37 (-1.50 to 0.00) and axis (degrees, °) 111.44 ± 70.08 (5.00 to 180.00).”

#RV4: The article just involved a single type of contact lens, which have different water content leading to diverse tear film dynamics for subjects. It is suggested to add more types to improve the feasibility and scientificity of the evaluation method.

#AU4: Thank you for taking the time to review our article. We appreciate your thoughtful comment regarding the inclusion of additional contact lens types in our study.

We understand the importance of comparing and integrating different types of contact lens materials based on water gradient, and it is our intention to do so in future studies. However, in this particular study, our aim was to analyze the tear film dynamics of a new monthly replacement contact lens material with a water gradient.

We would like to clarify that the contact lens we evaluated is the only one of its kind in the industry, and there is currently no comparator available. We believe that this new material is an important development in the field of contact lenses and warrants careful analysis and evaluation.

That being said, we acknowledge the significance of comparing different types of contact lenses in future studies to further improve the feasibility and scientific validity of our evaluation method. We appreciate your suggestion and will take it into consideration for our future research.

#RV5: The study only included a single follow-up time point of 30 days, which may not capture the full extent of tear film dynamics over a longer period.

#AU5: Thank you for your valuable feedback on our study. We appreciate your concern about the limited follow-up time point of 30 days.

We agree that a longer follow-up period would be valuable to capture the full extent of tear film dynamics over time. However, in this particular study, we chose a follow-up period of 30 days based on the nature of the contact lens material we were evaluating.

The contact lens in our study is a monthly replacement contact lens material, and therefore, it was important to evaluate the tear film dynamics of the contact lens after a full month of wear. This allowed us to observe the full effect of the contact lens on the tear film and evaluate its potential for improving dry eye disease symptoms.

That being said, we are currently working on a future study that will include a longer follow-up period of 6 months to evaluate the long-term effects of this contact lens material on tear film dynamics and dry eye disease symptoms. We believe that this will provide valuable insights into the sustained efficacy of the contact lens over a longer period of time.

Thank you again for your feedback, which we will take into consideration in our future studies.

References

  1. Lira, M.; Pereira, C.; Real Oliveira, M.E.C.D.; Castanheira, E.M.S. Importance of contact lens power and thickness in oxygen transmissibility. Cont. Lens Anterior Eye 2015, 38, 120–126, doi:10.1016/J.CLAE.2014.12.002.

Round 2

Reviewer 2 Report

The authors have improved the manuscript and this revised version can be acceptable.